# Characterization of a novel compound that promotes myogenesis *via* Akt and transcriptional co-activator with PDZ-binding motif (TAZ) in mouse C2C12 cells

Manami Kodaka[1�], Fengju Mao[1�], Kyoko Arimoto-Matsuzaki[1]*, Masami Kitamura[1,2], Xiaoyin Xu[1,3], Zeyu Yang[1,4], Kentaro Nakagawa[1], Junichi Maruyama[1], Kana Ishii[5], Chihiro Akazawa[5], Takuya Oyaizu[6,7], Naoki Yamamoto[6,7], Mari Ishigami-Yuasa[8], Nozomi Tsuemoto[9], Shigeru Ito[9], Hiroyuki Kagechika[8,9], Hiroshi Nishina[10], Yutaka Hata[1,11]*

1 Department of Medical Biochemistry, Graduate School of Medical and Dental Sciences, Tokyo Medical and Dental University, Tokyo, Japan, 2 Office for Gender Equality and Work-Life Balance/Office for Child Care Support, Tokyo Medical and Dental University, Tokyo, Japan, 3 Department of Breast Oncology Surgery, the Second Affiliated Hospital of Wenzhou Medical University, Wenzhou, China, 4 Department of Ultrasound, Shengjing Hospital of China Medical University, Shenyang, China, 5 Department of Biochemistry and Biophysics, Graduate School of Health Care Sciences, Tokyo Medical and Dental University, Tokyo, Japan, 6 Department of Orthopaedic and Spinal Surgery, Graduate School, Tokyo Medical and Dental University, Bunkyo-ku, Tokyo, Japan, 7 Hyperbaric Medical Center, Tokyo Medical and Dental University, Bunkyo-ku, Tokyo, Japan, 8 Chemical Biology Screening Center, Institute of Biomaterials and Bioengineering, Tokyo Medical and Dental University, Tokyo, Japan, 9 Institute of Biomaterials and Bioengineering, Tokyo Medical and Dental University, Tokyo, Japan, 10 Department of Developmental and Regenerative Biology, Medical Research Institute, Tokyo Medical and Dental University, Tokyo, Japan, 11 Center for Brain Integration Research, Tokyo Medical and Dental University, Tokyo, Japan

‚ These authors contributed equally to this work.
* kmatsuzaki.mbc@tmd.ac.jp (KA-M); yuhammch@tmd.ac.jp (YH)

**Data Availability Statement:** All relevant data are within the manuscript and its Supporting Information files.

## Abstract

Transcriptional co-activator with PDZ-binding motif (TAZ) plays versatile roles in the regulation of cell proliferation and differentiation. TAZ activity changes in response to the cellular environment such as mechanic and nutritional stimuli, osmolarity, and hypoxia. To understand the physiological roles of TAZ, chemical compounds that activate TAZ in cells are useful as experimental reagents. Kaempferol, TM-25659, and ethacridine are reported as TAZ activators. However, as each TAZ activator has a distinct property in cellular functions, additional TAZ activators are awaiting. We screened for TAZ activators and previously reported IB008738 as a TAZ activator that promotes myogenesis in C2C12 cells. In this study, we have characterized IBS004735 that was obtained in the same screening. IBS004735 also promotes myogenesis in C2C12 cells, but is not similar to IBS008738 in the structure. IBS004735 activates TAZ *via* Akt and has no effect on TAZ phosphorylation, which is the well-described key modification to regulate TAZ activity. Thus, we introduce IBS004735 as a novel TAZ activator that regulates TAZ in a yet unidentified mechanism.

**Funding:** YH; Grant Number 22590267, Japan Society for the Promotion of Science (https://www.jsps.go.jp/) YH; Grant Number 17im0210612h0001 and 18im0210612h0002, Japan Agency for Medical Research and Development (https://www.amed.go.jp/en/) K A-M; Suzuken Memorial Foundation (https://www.suzukenzaidan.or.jp/) K A-M; The Nakatomi Foundation (https://www.nakatomi.or.jp/) K A-M; Hoyu Science Foundation (https://www.hoyu.co.jp/zaidan/subsidy/) K A-M; Takeda Science Foundation (https://www.takeda-sci.or.jp/business/assist.html) K A-M; The Nakajima Foundation (http://www.nakajimafound.or.jp/) K A-M; Kobayashi Foundation (https://www.kisf.or.jp/english/) K A-M; The Japan Foundation for Aging and Health (https://www.tyojyu.or.jp/en/index.html) The funders had no role in study design, data collection and analysis, decision to publish, or preparation of the manuscript.

**Competing interests:** Yutaka Hata collaborated with Shinogi Co. Ltd. to develop drugs against sarcopenia between 2017 and 2018. This does not alter our adherence to PLOS ONE policies on sharing data and materials.

## Introduction

Transcriptional co-activator with PDZ-binding motif (TAZ) shuttles between the cytoplasm and the nucleus [1]. TAZ interacts with various transcription factors inside the nucleus and regulates versatile genes. TAZ is phosphorylated by large tumor suppressor (LATS) kinases, the core kinases of the Hippo pathway. Phosphorylation generates 14-3-3-binding motif. Consequently, TAZ is segregated in the cytoplasm. Phosphorylation also triggers TAZ degradation. In this way, the tumor suppressor Hippo pathway negatively regulates TAZ [2]. In cancer cells, dysregulation of the Hippo pathway leads to hyperactivation of TAZ. Active TAZ cooperates with TEA-domain (TEAD) family members to induce epithelial-mesenchymal transition (EMT) and enhances drug resistance [3, 4]. TAZ cross-talks with WNT pathway and confers cancer stemness [5]. In mesenchymal stem cells, TAZ promotes myogenesis and osteogenesis, and inhibits adipogenesis [6]. TAZ is required for lung alveolar cell differentiation and heart development [7–11]. TAZ promotes bone formation and suppresses chondrogenesis [12–15]. TAZ maintains testicular function in aged mice [16]. To study the physiological and pathophysiological roles of TAZ, loss-of-function and gain-of-function approaches are frequently used in animals. Knockout animals are the most straight forward tools to reveal essential roles of TAZ. To evaluate the effect of TAZ hyperactivation, TAZ mutants, which lack LATS-phosphorylation site(s) and are constitutively active, are enforcedly expressed. Alternatively, the suppression of components of the Hippo pathway (for examples, mammalian Ste20-like kinases, salvador and Mobs), is adopted [17–19]. Likewise, knockdown and knockout approaches and expression of TAZ active mutants are common strategies for the analysis at the cell level. However, these methods are not appropriate to study the relatively short-term or acute effect of TAZ inactivation or activation. To this end, reagents to inhibit and activate TAZ are essential. Verteporfin, although it was originally developed as a photosensitizer for photodynamic therapy, is the best characterized inhibitor and is widely used as an experimental reagent [20]. On the other hand, several TAZ activators are reported. Kaempferol and TM-25659 promote osteogenesis in C3H10T1/2 and human adipose-derived stem cells and inhibits adipogenesis in 3T3-L1 cells [21, 22]. Ethacridine inhibits adipogenesis in C3H10T1/2 cells and induces thyroid follicular cell differentiation form human embryonic stem cells [23, 24]. IBS008738 facilitates myogenesis in C2C12 cells [25]. Although all these compounds are commercially available, TAZ activators are not yet fully established. Therefore, it is meaningful to provide a novel TAZ activator to researchers.

We previously performed a cell-based assay to screen for TAZ activators by using MCF10A cells expressing TAZ (MCF10A-TAZ) [25]. We cultured MCF10A-TAZ cells in the serum-free medium supplemented with insulin, epithelial growth factor and basic fibroblast growth factor in the ultra-low attachment plate. When large tumor suppressor kinase 1 and -2 (LATS1/2) are suppressed to activate TAZ, cells form spheres. *LATS1/2* silencing has no effect in parent MCF10A cells without overexpressed TAZ, while *TAZ* silencing inhibits sphere formation in MCF10A-TAZ cells. It means that the sphere formation depends on the activity of TAZ. Therefore, we can regard the compounds that enable MCF10A-TAZ cells to form spheres as TAZ activators. We applied 18,459 small chemical compounds to MCF10A-TAZ cells and obtained 50 compounds that induced the sphere formation (S1A Fig and S2 Fig). These compounds also enhanced TAZ-TEAD reporter activity in HEK293FT cells (S1B Fig). We applied these compounds to mouse myoblast C2C12 cells and found 43 compounds that enhanced myogenesis (S1C Fig). Among them, four compounds (FKL01303, IBS000145, IBS004735, and IBS008738) strongly promoted myogenesis in mouse myoblast C2C12 cells (S1C Fig, arrows). FKL01303 is 1-[5-hydroxy-1-(4-methoxyphenyl)-2-methylindol-3-yl]ethenone (Amendol). Amendol is reported to activate sphingosine-1-phosphate receptor 1 (SPR1) (https://pubchem.

ncbi.nlm.nih.gov/compound/658914). Therefore, FKL01303 may activate TAZ through SPR1 [26]. We focused on three remaining uncharacterized compounds. In the previous study, we characterized IBS008738 and reported it as a TAZ activato that promotes skeletal muscle repair and prevents dexamethasone-induced muscle atrophy [25]. In this study, we have focused on IBS004735, while the property of IBS000145 will be reported in future.

## Results

### IBS004735 enhances protein expressions of myogenic differentiation and its effect depends on TAZ

IBS004735 is structurally distinct from other TAZ activators (Fig 1A). IBS004735 promoted myofusion (Fig 1B) and enhanced the expression of myosin heavy chain (MHC) in C2C12 cells (Fig 1C). The immunoblottings demonstrated the enhanced expression of MHC and myogenin at 24, 48, and 72 h (Fig 1D). MyoD expression and TAZ expression were also slightly enhanced. TAZ knockdown abolished the effect of IBS004735 on MHC expression, supporting that the myogenesis-promoting activity depends on TAZ (Fig 1E). We also confirmed that IBS004735 had no effect on the proliferation of C2C12 cells under the growing condition (Fig 1F).

### IBS004735 enhances mRNAs of myogenic markers

In quantitative real-time PCR (qRT-PCR), the gene transcriptions of myogenin (*Myog*) and MyoD (*Myod1*) at 24 h under the differentiation condition were enhanced in IBS004735-treated cells (Fig 2A). Although TAZ protein expression was enhanced at 48 and 72 h (Fig 1D), TAZ mRNA (*Wwtr1*) did not increase at 24 h and rather decreased at 72 h, suggesting that TAZ expression was changed at the protein level. No significant increase was observed at 24 h in mRNAs of connective tissue growth factor (*Ctgf*) and cyclin D1(*Ccnd1*), which are regarded as TEAD target genes in epithelial cells. After C2C12 cells were cultured for 72 h, *Myog* mRNA was still higher in IBS004735-treated cells, while the difference was lost for *Myod1*. At 72 h, *Ctgf* expression was enhanced in IBS004735-treated cells.

### IBS004735 enhances MyoD-responsive myogenin promoter reporter in C2C12 cells

We tested whether and how IBS004735 affects the reporter activities mediated by various TAZ-interacting transcription factors in C2C12 cells. We exogenously expressed each reporter and TAZ in C2C12 cells and treated with DMSO or 10 μM IBS004735. TAZ overexpression enhanced the activities of MyoD-, TEAD-, and Pax3-reporters (Fig 2B, the first and the third columns). TAZ also showed the tendency to augment SMAD-reporter. IBS004735 further enhanced MyoD- and TEAD-reporters in C2C12 cells (Fig 2B, the third and fourth columns). In this regard, IBS004735 is different from IBS008738, which does not affect TEAD-reporter in C2C12 cells. IBS004735 did not enhance the effect of TAZ on Pax3- or SMAD- reporter. In Fig 2A, IBS004735 did not increase *Ctgf* mRNA at 24 h, but in Fig 2B, it enhanced TEAD-reporter after 18 h-treatment. These results are apparently inconsistent. For the reporter assay, we used the luciferase reporter driven by eight repeats of the TEAD-responsive sequence. We suspect that this artificial construct may respond to TEAD more strongly than the native *Ctgf* promoter.

### IBS004735 increases MyoD association with the myogenin promoter

MyoD, TEAD4, and Pax3 were immunoprecipitated from differentiated C2C12 cells treated with control DMSO or IBS004735 and chromatin immunoprecipitation assays were

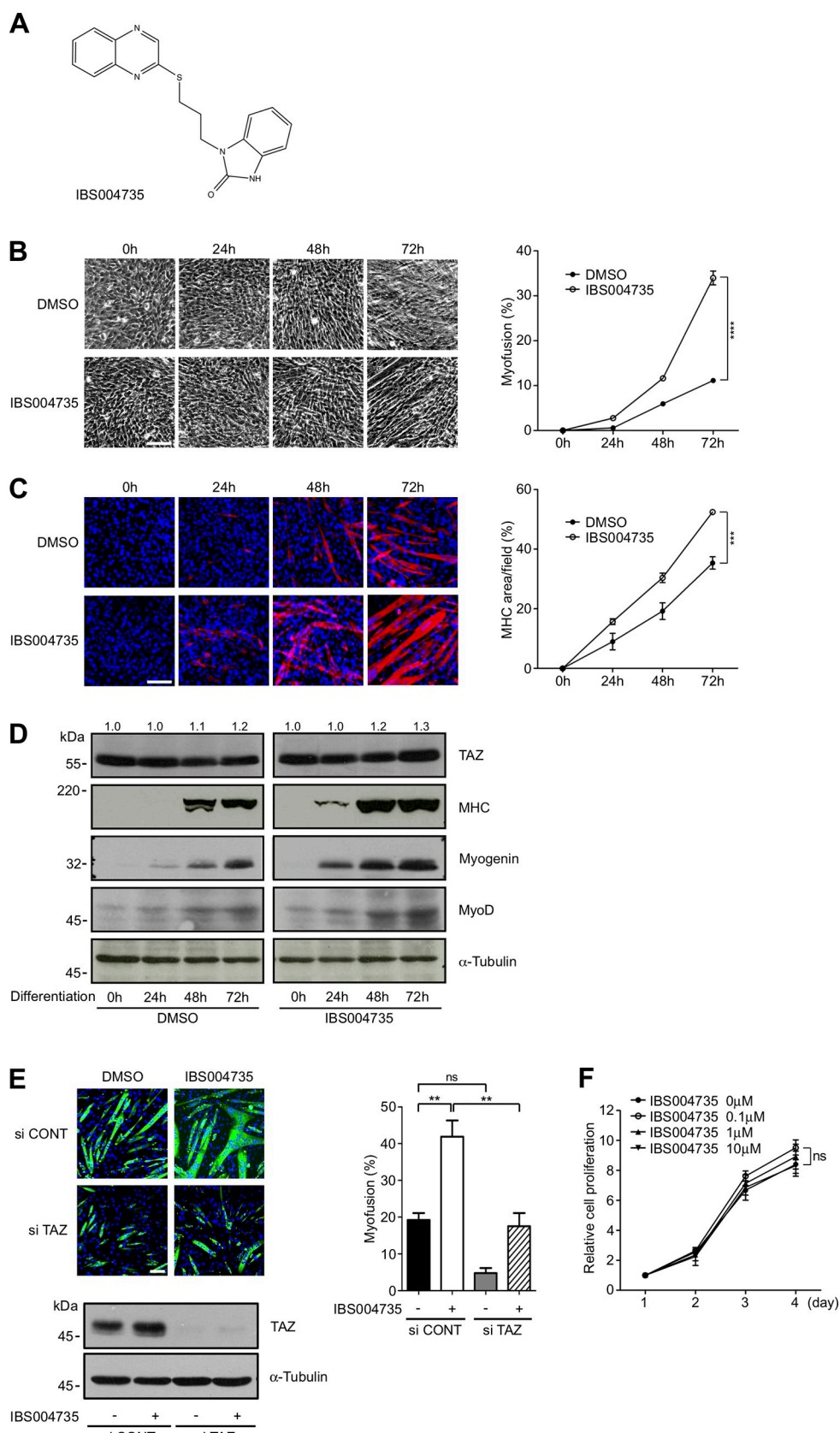

**Fig 1. The effect of IBS004735 on myogenesis and myofusion in mouse C2C12 cells.** (A) Chemical structure of IBS004735. (B), (C), and (D) C2C12 cells were cultured in differentiation medium with either control DMSO or 10 μM IBS004735. In (B), myofusion was promoted by IBS004735. Scale bar, 100 μm. Myofusion index was calculated as described in Methods. Data are means and standard errors of the means. ****, p<0.0001. In (C), the cells were immunostained with anti-myosin heavy chain (MHC) antibody and the signals were measured by ImageJ. ***, p<0.001. In (D), the whole cell lysates were immunoblotted with the indicated antibodies. 50 μg of total protein is charged in each lane and the immunoblotting with anti-α-tubulin antibody is shown as a loading control. Signals of TAZ and α-tubulin were quantified by ImageJ. Numbers indicate relative TAZ signals normalized by α-tubulin signals. The value at 0 h was set at 1.0 (E) C2C12 cells were transfected with either control siRNA (si CONT) or TAZ siRNA (si TAZ). 24 h later, the cells were replated and cultured in differentiation medium for 72 h with DMSO or 10 μM IBS004735. MHC was immunostained (green). Nuclei were visualized with Hoechst 33342 (blue). Cell lysates were immunoblotted with anti-TAZ and anti-α-tubulin antibodies to show that TAZ was indeed suppressed. (F) C2C12 cells were plated at 1,000 cells/well in a 96-well plate and cultured in the growing medium with various doses of IBS004735. MTT assay was performed at the indicated time points.

performed. Consistent with the reporter assays, IBS004735 increased MyoD binding to *Myog* promoter, but had no effect on Pax3 binding to *Myf5* promoter (Fig 2C). The association of TEAD4 with *Ctgf* promoter was not affected. This result is consistent with that of qRT-PCR (Fig 2A), but not with that of the reporter assay (Fig 2B). We consider that this discrepancy can also be explained by the difference between the native *Ctgf* promoter and the artificial promoter.

## IBS004735 does not affect the subcellular localization of TAZ and the phosphorylation at Serine 89

In the canonical Hippo pathway, TAZ is phosphorylated by LATS kinases and negatively regulated [27]. The phosphorylated TAZ is segregated in the cytoplasm and undergoes degradation, while the unphosphorylated TAZ is accumulated in the nucleus and regulates gene transcription. We examined whether IBS004735 increases the unphosphorylated nuclear TAZ in C2C12 cells. In the subcellular fractionation, TAZ was distributed in the cytoplasmic and nuclear fractions in C2C12 cells, and IBS004735 did not change the distribution of TAZ (Fig 3A). TAZ is phosphorylated by LATS kinases at four sites. Among them Serine 89 is the most important, because the phosphorylation at this site generates the 14-3-3-binding site. We analyzed the whole cell lysates of C2C12 cells treated with IB004735 on a Phos-tag gel (Fig 3B). The top band was detected with both anti-TAZ and anti-phospho-Serine 89 (p-TAZ) antibodies (Fig 3B, an arrow), while the lower two bands were not detected with anti-phospho-Serine 89 antibody (Fig 3B, arrowheads). Ethacridine, which is known to promote dephosphorylation of TAZ, increased not only the expression of whole TAZ but also that of unphosphorylated TAZ (Fig 3B, a white arrowhead) [23]. In contrast, IBS004735 did not increase unphosphorylated TAZ expression. IBS004735 is also different from IBS008738, which increases unphosphorylated TAZ [25]. The coimmunoprecipitation experiment demonstrated that IBS004735 did not augment the interaction between TAZ and TEAD4 in HEK293FT cells (**Fig 3C**). These findings suggest that the activation of TAZ by IBS004735 is not mediated by the suppression of the canonical Hippo pathway.

## IBS004735 activates Akt-mTOR-S6K pathway and stimulates protein synthesis in C2C12 cells

We attempted to clarify the mechanism, by which IBS004735 activates TAZ. Insulin-like growth factor 1-Akt axis plays an important role in the regulation of protein synthesis and protein degradation in skeletal muscles. We therefore examined the effect of IBS004735 on Akt signaling pathway. IBS004735 stimulated Akt phosphorylation (Fig 4A). mTOR and S6K were

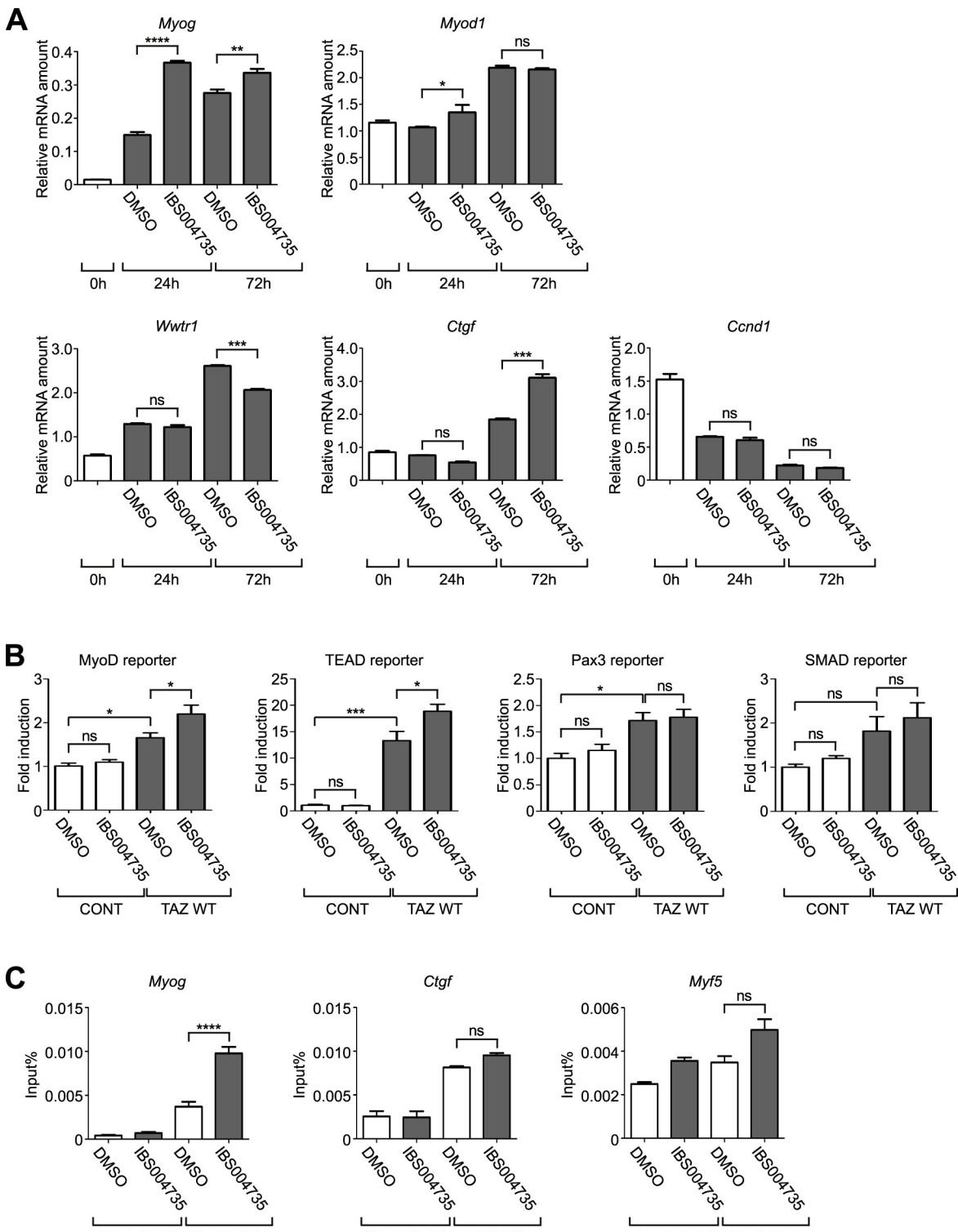

**Fig 2. The effect of IBS004735 on the expression of myogenic marker genes.** (A) qRT-PCR analysis of indicated genes. After C2C12 cells were transferred from growth medium to differentiation medium, mRNAs were collected at the indicated time points and qRT-PCRs were performed. IBS004735 enhanced the expression of myogenin (*Myog*) and MyoD (*Myod1*) at 24 h. Connective tissue growth factor (*Ctgf*) expression was enhanced at 72 h by IBS004735. *Ccnd1* expression was not affected. Data are means and standard errors of the means. ns, not significant; **, p<0.01; and ***, p<0.001. (B) C2C12 cells were transfected pGL3 Myo-184- (for MyoD), 8xGT-IIC-d51LucII (for TEAD), 9xCAGA-MLP (for SMAD), and p(PRS-1/-4)x3 (for Pax3) luciferase reporters alone (CONT) or with TAZ (TAZ

WT). pCMV alkaline phosphatase was used as a control. The cells were transferred to differentiation medium 24 h later and cultured for another 18 h with DMSO or 10 μM IBS004735. MyoD-, TEAD-, and Pax3- reporter activities were enhanced by TAZ. IBS004735 further enhanced MyoD- and TEAD-reporters, but had no effect on Pax3- or SMAD-reporter. Data are means and standard errors of the means. ns, not significant; *, p<0.05; and ***, p<0.001. (C) C2C12 cells that were cultured with DMSO or 10 μM IBS004735 for 24 h in differentiation medium. Chromatin immunoprecipitation assays were performed with anti-MyoD, anti-TEAD, and anti-Pax3 antibodies. The co-precipitated promoters of *Myog*, *Ctgf*, and *Myf5* were detected by PCR. Protein G Sepharose 4 Fast Flow beads were used as a control (Mock ChIP). IBS004735 promoted the association of MyoD to the myogenin promoter. Data are means and standard errors of the means. ns, not significant; and ****, p<0.0001.

also activated. We next evaluated the effect of IBS004735 on protein synthesis at different time points in the SUnSET experiment. Consistent with the up-regulation of Akt-mTOR-S6K axis, the incorporation of puromycin was higher in IBS004735-treated cells at 0 h and 24 h, although the difference was less remarkable at 48 and 72 h (Fig 4B). The inhibition of Akt by Akti-1/2 or *Akt1/2* silencing attenuated the effect of IBS004735 (Fig 4C and 4D). Likewise, the inhibition of mTOR by rapamycin significantly compromised the effect of IBS004735 (Fig 4E). These findings indicate that IBS004735 activates Akt-mTOR-S6K pathway and stimulates protein synthesis in C2C12 cells, especially at the early differentiation phase. Interestingly, in the TAZ-negative background, Akt phosphorylation was still enhanced by IBS004735 at 24 h (Fig 4F, si TAZ, arrowheads), but the IBS004735-mediated enhancement was reduced at 48 h (Fig 4F, si TAZ, arrows). These findings imply that IBS004735 initially activates TAZ *via* Akt signaling, and that TAZ subsequently activates Akt.

## IBS004735 activates Akt through tyrosine kinase and phosphoinositide 3-kinase (PI3K)

The next obvious question is the mechanism, by which IBS004735 activates Akt-mTOR-S6K pathway. When C2C12 cells were pre-treated with PI3K inhibitor, LY294002, the activation of Akt was abolished and the effect of IBS004735 was significantly compromised (Fig 5A). Genistein, a tyrosine kinase inhibitor, also abolished the effect of IBS004735 (Fig 5B). These findings suggest that a certain tyrosine kinase, which is coupled with PI3K, may be involved in the function of IBS004735.

## IBS004735 blocks the inhibitory effect of tumor necrosis factor- (TNF-) and myostatin

TNF-α and myostatin play important roles in muscle atrophy associated with cancer cachexia and various chronic diseases [28, 29]. IBS008738 antagonized the inhibitory effect of myostatin in C2C12 cells. We tested whether IBS004735 antagonizes TNF-αand myostatin. 10 ng/ml TNF-α and 100 ng/ml myostatin suppressed MHC expression and myofusion in C2C12 cells but 10 μM of IBS004735 partially but significantly recovered them (Fig 6A and 6B). We further examined whether and how IBS004735 affects the signals of TNF-α and myostatin. In the reporter assays using HEK293FT cells. TNF-α activated NFκB-reporter activity, while IBS004735 decreased it (Fig 6C). Likewise, IBS004735 attenuated myostatin-induced SMAD-reporter activity (Fig 6D).

## IBS004735 suppresses the expression of E3 ligases

E3 ligases, Trim63 (Murf1), Fbxo30 (Musa1), and Fbxo32 (Atrogin-1), are involved in muscle atrophy [30, 31]. IBS008378 suppressed dexamethasone-induced enhancement of *Trim63* and *Fbxo32* in C2C12 cells. We examined the effect of IBS004735 on the expression of these genes (Fig 6E). *Trim63* and *Fbxo32* were suppressed by IBS004735. *Fbxo30* was also reduced,

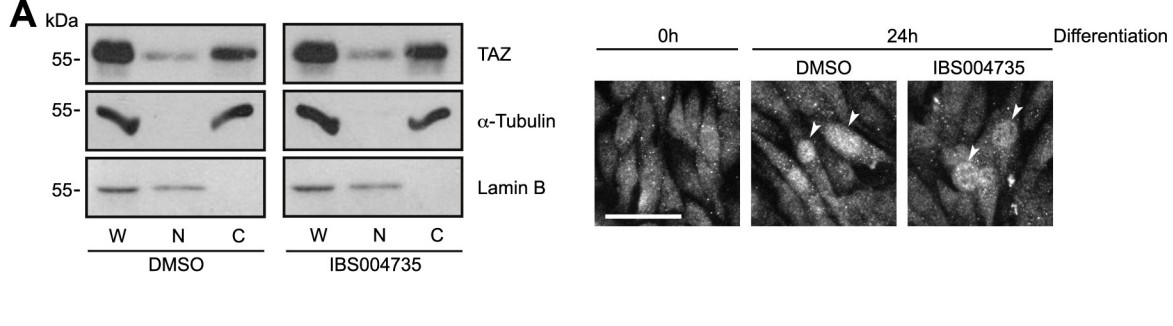

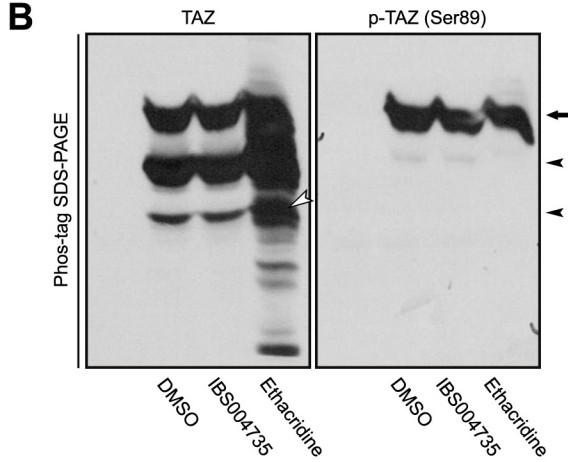

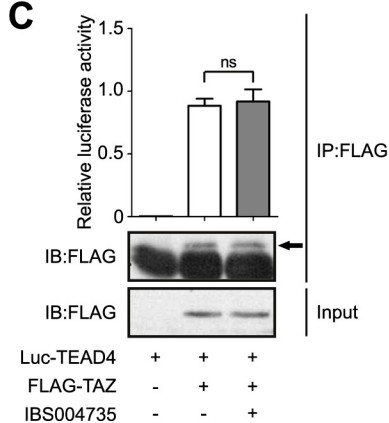

**Fig 3. The effect of IBS004735 on the subcellular localization and phosphorylation of TAZ and on the interaction between TAZ and TEAD4.** C2C12 cells were cultured in differentiation medium for 24 h and harvested. In (A), the subcellular fractionation was performed. The comparable amounts of the whole cell lysates (W), the nuclear fraction (N), and the cytoplasmic fraction (C) were immunoblotted with the indicated antibodies. α-Tubulin and lamin B were used as the cytoplasmic marker and the nuclear marker, respectively. TAZ was detected in the nuclear and cytoplasmic fractions and IBS004735 did not change the distribution. TAZ migrated more slowly in the cytoplasmic fraction than in the nuclear fraction, which suggests that the cytoplasmic TAZ is phosphorylated. In the right, the cells were immunostained by anti-TAZ antibody. After differentiation, TAZ distribution was heterogenous in C2C12 cells. In some cells, TAZ was accumulated in the nuclei (arrowheads). Scale bar, 100 μm. In (B), HEK293FT cells were transfected with pCIneoGFP-TAZ. The whole cell lysates were run on Phos-tag gels and immunoblotted with anti-TAZ (left, TAZ) and anti-phospho-Serine 89 antibody (right, p-TAZ). The top bands were detected with both antibodies (an arrow), while the lower two bands were detected with only anti-TAZ antibody (arrowheads). As TAZ has multiple phosphorylation sites, the middle band may correspond to TAZ that is phosphorylated at other sites than Serine 89. The lysates of the cells treated by 10 μM ethacridine were used as a control. Unphosphorylated TAZ (a white arrowhead) was remarkably increased by ethacridine. (C) HEK293FT cells were transfected with pCIneoLuc-TEAD4 and pCIneoFH-TAZ. 24 h later, the cells were treated with either DMSO (a white bar) or 10 μM IBS004735 (a gray bar) for 24 h. The immunoprecipitation was performed with anti-DYKDDDDK-tag beads. The luciferase activity attached to the beads were measured.

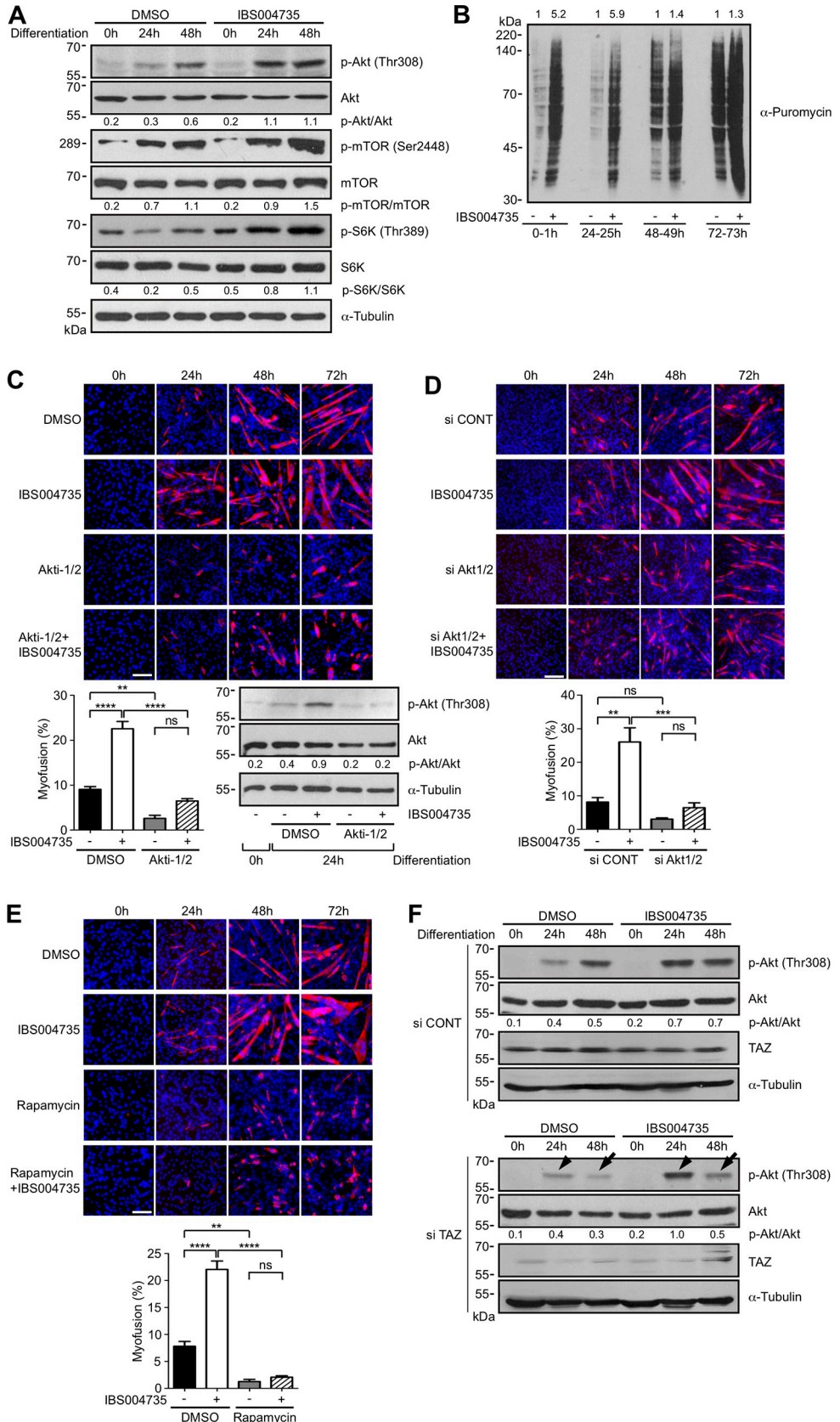

**Fig 4. IBS004735 activates Akt-mTOR-S6K axis.** (A) C2C12 cells were cultured in differentiation medium with DMSO or 10 μM IBS004735 for the indicated periods of time. The cell lysates were immunoblotted with the indicated antibodies. IBS004735 increased phosphorylation of Akt, mTOR, and S6K at indicated sites. Signals were quantified by ImageJ. Numbers indicate relative intensities of phospho-Akt, -mTOR, and -S6K signals normalized by total-Akt, -mTOR, and -S6K signals. (B) C2C12 cells were cultured under the differentiation condition with DMSO or 10 μM IBS004735 for the indicated periods of time. 1 μM puromycin was added to the medium. 30 min later, the cells were harvested and the incorporated puromycin was detected with anti-puromycin antibody. At 0 h and 24 h, IBS004735 enhanced puromycin incorporation. At 48 h and 72 h, the effect of IBS004735 was less remarkable. Signals were quantified by ImageJ. Numbers indicate relative intensities of signals in IBS004735-treated cells normalized signals in control cells at each time point. (C), (D), and (E) C2C12 cells were cultured in differentiation medium containing the indicated reagents (DMSO, 1 μM Akti-1/2, 1 nM rapamycin, and 10 μM IBS004735). The cells were immunostained with anti-MHC antibody. Nuclei were visualized with Hoechst 33342. Myofusion index was calculated. Data are means and standard errors of the means. **, p<0.01; ***, p<0.001; and ****, p<0.0001. In (D), first, *Akt1/2* were knocked down by siRNAs against mouse *Akt1/2*. 24 h later, the cells were transferred to the differentiation medium with or without 10 μM IBS004735. (F) C2C12 cells were transfected with control siRNA (si CONT) or TAZ siRNA (si TAZ) and then cultured under the differentiation condition with DMSO or 10 μM IBS004735. Although TAZ knockdown remarkably suppressed TAZ expression in DMSO-treated cells, TAZ was slightly expressed with IBS004735 treatment. TAZ depletion did not change IBS004735-induced enhancement of phospho-Akt at 24 h (arrowheads) but attenuated it at 48 h (arrows). In (C) and (D), number indicate the relative ratios of p-Akt signals normalized by Akt signals.

although the difference was not statistically significant. The role of autophagy in muscle atrophy is debatable. FoxO3 stimulates autophagic/lysosomal pathway and induces protein degradation in skeletal muscles [30, 31]. On the other hand, in *Drosophila*, autophagy prevents the accumulation of protein aggregates in muscles and delays muscle ageing [32]. In mouse, autophagy is dysregulated and p62 is accumulated in old muscles [33, 34]. These findings imply that autophagy plays a role in homeostasis of skeletal muscles. Therefore, we examined the effect of IBS004735 on molecules that were involved in autophagy. However, IBS004735 did not show any effect on the expression of p62, LC3, Atg5, and cathepsin L (Fig 6F). AMP-activated protein kinase-mediated phosphorylation site (Serine 555) of ULK1 was reduced under differentiation but IBS004735 did not show any effect.

## IBS004735 promotes myogenesis in primary cultured human myoblasts

To examine whether IBS004735 is active not only for mouse cells but also for human cells, we applied the compound to primary cultured human myoblasts. IBS004735 increased myofusion index and MHC expression (Fig 7A). In the immunoblotting, IBS004735 facilitated MHC expression at 24 h and enhanced myogenin expression at 24, 48, and 72 h (Fig 7B). All these findings support that IBS004735 is active for human cells.

## The effect of derivatives of IBS004735

To obtain a compound with higher activity and gain insight into which residues are essential for the activity, we generated 21 derivatives (S3 Fig and S4A Fig). However, among them, only KQ-8 facilitated myofusion and MHC expression (S4B and S4C Fig). Therefore, the whole structure of IBS004735 might be important for its activity.

## Discussion

The Hippo pathway was initially identified as the signaling pathway that regulates organ size in *Drosophila melanogaster* [35]. In human cancers, the Hippo pathway is frequently dysregulated [36]. Consequently, TAZ is activated and causes metastasis and recurrence. Clinical data indicate that cancer patients with TAZ activation exhibit poor prognosis. On the other hand, TAZ plays important roles in tissue stem cells [37]. TAZ is necessary for normal organogenesis in heart, lung, and tooth and for tissue repair in skin and heart [7, 9, 11, 38]. TAZ also contributes to myogenesis and osteogenesis [6]. Therefore, TAZ attracts attention in the field of

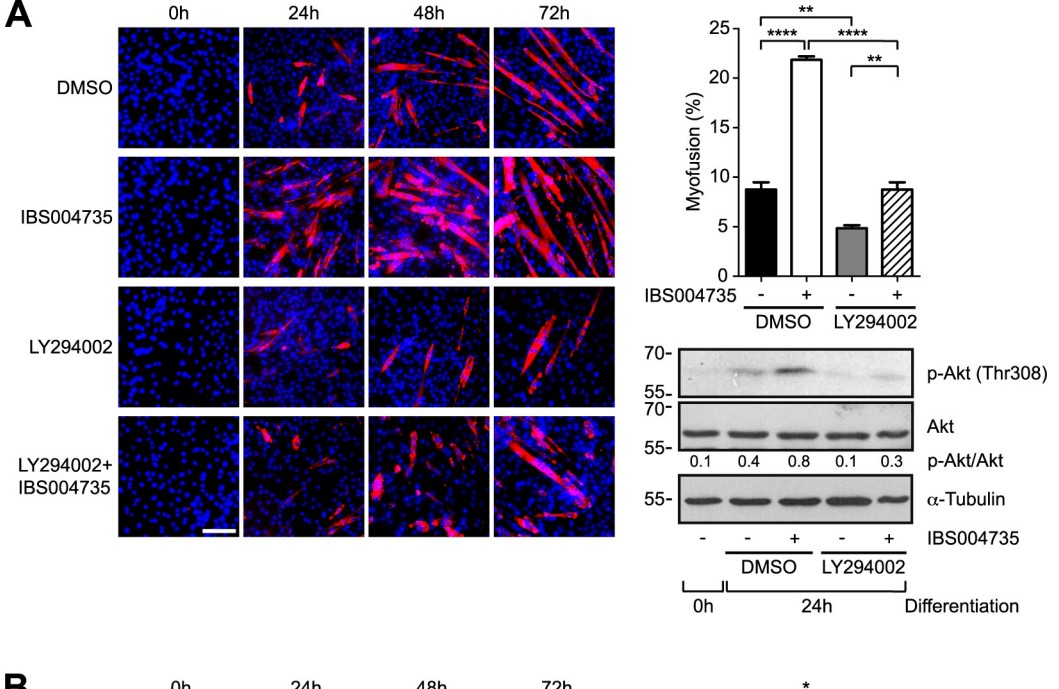

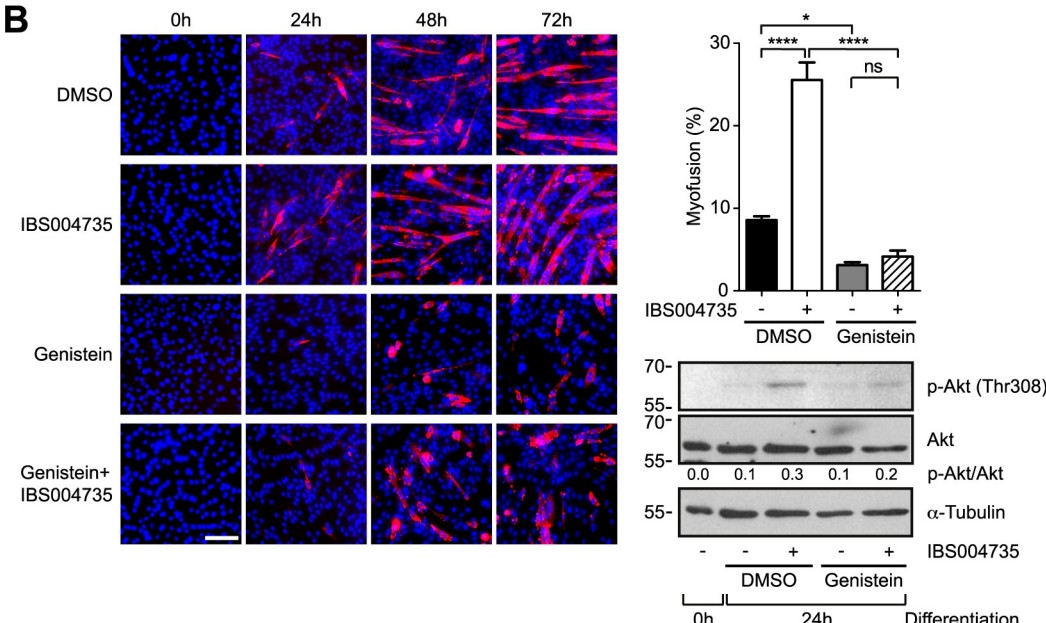

**Fig 5. IBS004735 initially activates Akt independently of TAZ and the tyrosine kinase inhibitor and PI3K inhibitors abolish the effect of IBS004735.** C2C12 cells were cultured in differentiation medium containing indicated reagents. (DMSO, 1 μM LY294002 in (A), 20 μM genistein in (B), and 10 μM IBS004735). The cells were immunostained with anti-MHC antibody. Nuclei were visualized with Hoechst 33342. Myofusion index was calculated. Data are means and standard errors of the means. *, p<0.05; **, p<0.01; ***, p<0.001; and ****, p<0.0001. The immunoblottings demonstrate that LY294002 and genistein attenuated the IBS004735-mediated enhancement of Akt phosphorylation. The signals of p-Akt and Akt were quantified and the ratios between two signals were indicated.

cancer biology and regenerative medicine [39]. However, TAZ is also known as a mechanosensor [40]. The Hippo pathway responds to nutrition signal, osmolarity, and hypoxia [41, 42]. YAP1, a TAZ homologue, is activated by hypoxia [43]. TEAD, the most prominent TAZ-

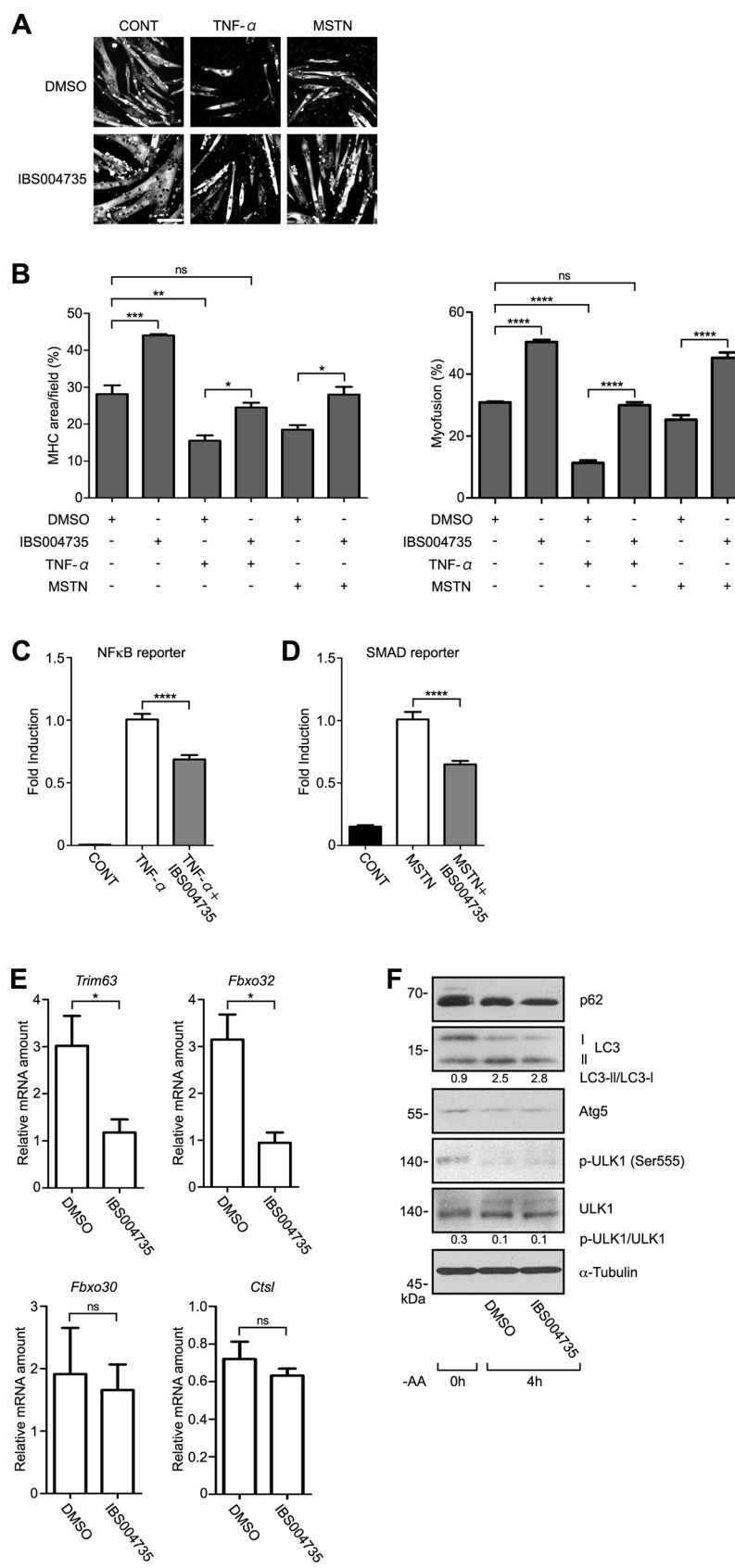

**Fig 6. The antagonistic effect of IBS004735 against TNF-α and myostatin and the effect on the expression of E3 ligases and autophagy-related proteins.** (A) and (B) C2C12 cells were cultured in differentiation medium containing various combinations of DMSO, 10 ng/ml TNF-α, 100 ng/ml myostatin (MSTN), and 10 μM IBS004735. 72 h later, the cells were immunostained with anti-MHC antibody and the nuclei were visualized with Hoechst 33342. TNF-α and myostatin reduced MHC expression and myofusion but IBS004735 partially recovered them. Data are means and standard errors of the means. ns, not significant; ****, p<0.0001. (C) and (D) NF-κB- and SMAD-reporter assays were performed in HEK293FT cells as described in Materials and Methods. TNF-α and myostatin activate the reporter activities, but IBS004735 attenuated them. Data are means and standard errors of the means. ****, p<0.0001. (E) C2C12 cells were cultured for 24 h under the differentiation condition with or without 10 μM IBS004735. mRNAs for indicated genes were quantified by qRT-PCR. *Trim63* (*Murf1*) and *Fbxo32* (*Atrogin1*) were reduced by IBS004735 (*, p<0.05), while *Fbxo30* (*MUSA1*) or *Ctsl* (*Cathepsin L*) was not. (F) C2C12 cells were cultured for 4 h under the amino acid-deprived condition to induce autophagy with or without 10 μM IBS004735. The cells lysates were immunoblotted by the indicated antibodies. IBS004735 did not affect the expression of autophagy-related proteins. Numbers indicate the ratios between LC3-I and LC3-II signals and between pULK1 and total ULK1 signals.

binding partner, is regulated by osmotic stress [44]. TAZ activity may be regulated in response to the continuously changing cellular environment. To understand quick effects of TAZ activation and inactivation, TAZ inhibitors and activators are important. We previously reported

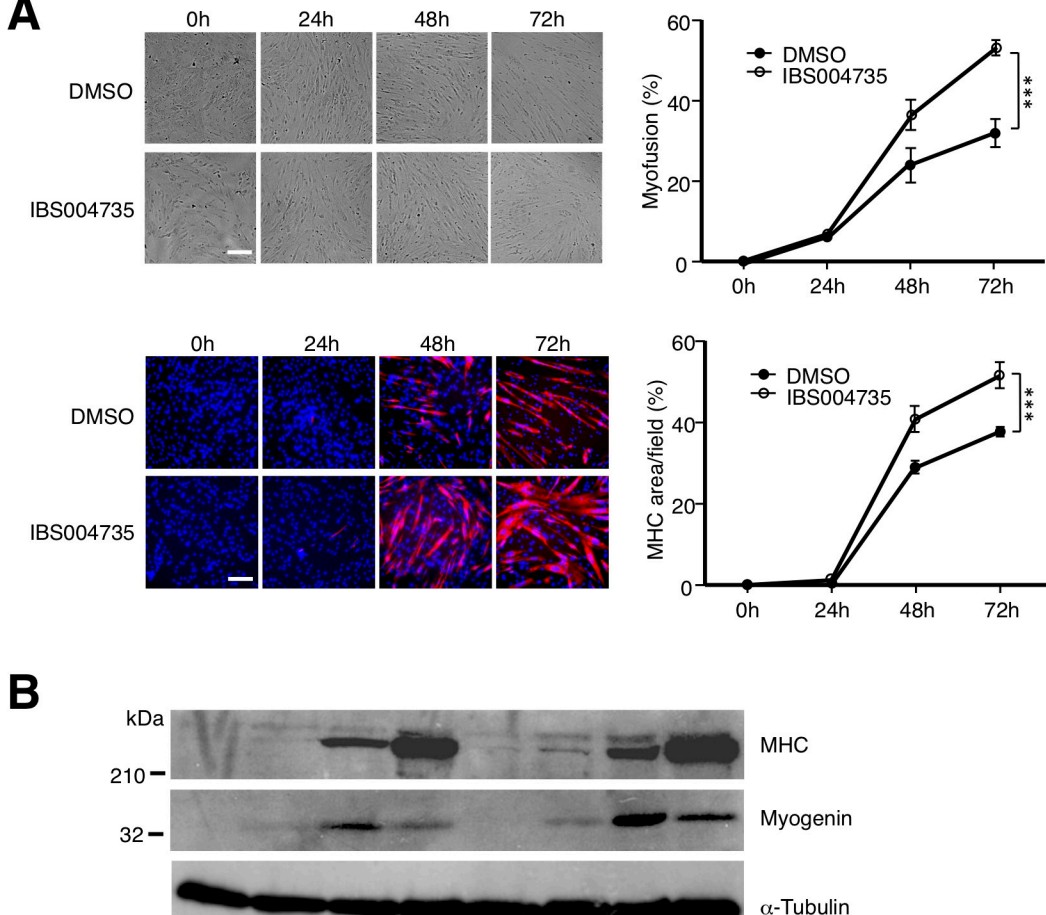

**Fig 7. IBS004735 promotes myogenesis in primary cultured human myoblasts.** Primary cultured human myoblasts were treated with 10 μM IBS004735 under the differentiation condition. Myogenesis was evaluated as described for Fig 1. (A) IBS004735 promoted myofusion and MHC expression. Scale bar, 100 μm. (B) IBS004735 enhanced MHC expression at 24 h and myogenin expression at 24, 48, and 72 h.

two TAZ activators, IBS008738 and ethacridine [23, 25]. In this study, we have characterized IBS004735 as a novel TAZ activator.

IBS004735 was obtained in the same screening that yielded IBS008738 [25]. TAZ activation allows immortalized human mammary epithelial MCF10A cells to survive in the floating condition and form spheres in the serum-free medium containing insulin, epidermal growth factor, and basic fibroblast growth factor. We identified 50 compounds that induced the sphere formation of TAZ-expressing MCF10A cells. They activate TEAD reporter in HEK293FT cells in the presence of TAZ. These findings support that the compounds indeed activate TAZ. However, it does not mean that the compounds directly bind TAZ. As shown in S1 Fig, all 50 compounds have distinct structures. Intriguingly, the cellular effects of compounds are not the same. Diversity of structures and properties suggest that each compound has a distinct target and activates TAZ through a different mechanism. Therefore, to study a wide variety of roles of TAZ, we consider that it is meaningful to characterize these TAZ activators.

IBS004735 facilitates myogenesis and myofusion in C2C12 cells (Fig 1). IBS004735 enhances myogenic markers such as myogenin and MHC. The association of MyoD to the myogenin promoter is also promoted (Fig 2). TAZ depletion abolishes the effect of IBS004735, supporting that IBS004735 requires TAZ to exhibit its effect (Fig 1). According to the model of the canonical Hippo pathway, TAZ activator is predicted to increase unphosphorylated nuclear TAZ. However, IBS004735 does not change the subcellular localization and the phosphorylation of TAZ in C2C12 cells (Fig 3). Hence, IBS004735 is distinct from IBS008738, which increases unphosphorylated TAZ. We speculate that IBS004735 activates TAZ independently of the canonical Hippo pathway.

IBS004735 activates Akt kinases (Fig 4). The inhibition of tyrosine kinase and PI3K and the suppression of Akt kinases abolish the effect of IBS004735, suggesting that IBS004735 activates TAZ *via* a certain tyrosine kinase coupled with PI3K and Akt kinases (Fig 5). Indeed, several receptor tyrosine kinases are found to activate YAP1 and TAZ [45]. IBS004735 may target of one of them. Akt is activated in the TAZ-negative background at 24 h after IBS004735 treatment, while the sustained activation of Akt by IBS004735 is attenuated by TAZ depletion. This finding suggests that IBS004735 initially activates Akt independently of TAZ but that TAZ is required to maintain Akt activation. TAZ up-regulates insulin receptor substrate 2 (IRS2), which recruits PI3K and activates Akt [46]. Therefore, it is not surprising that IBS004735-mediated TAZ activation maintains Akt activation. In contrast, the mechanism, by which the initial activation of Akt leads to TAZ activation, is not clear. Although there are several reports that Akt activates TAZ, the underlying molecular mechanism is yet unknown [47, 48]. One paper reported that Akt inhibits GSK3β and increases TAZ expression level [47]. Indeed, IBS004735 enhanced TAZ expression in C2C12 cells (Fig 1). Another paper reported that insulin receptor and G protein-coupled receptor cross-talk and activate YAP1 and TAZ in pancreatic cancer cells [49] and that the activation is cancelled by PI3K inhibitor. Although they did not show the requirement of Akt activation, there may be a certain mechanism by which PI3K activates YAP1 and TAZ and the target of IBS004735 could be placed in that pathway.

IBS004735 antagonizes the inhibitory effect of TNF-α on myogenesis in C2C12 cells (Fig 6). This finding is consistent with the myogenesis-promoting capacity of IBS004735. However, the underlying mechanism needs to be clarified, especially because YAP1, TAZ homologue, rather up-regulates NFκB signal in C2C12 cells [50]. Likewise, how TAZ blocks myostatin-mediated SMAD reporter is unclear. However, we could speculate that TAZ may trap SMAD proteins to interfere with myostatin signal.

In conclusion, IBS004735 is a novel TAZ activator that promotes myogenesis *via* Akt and TAZ in C2C12 cells. The identification of the target of IBS004735 will give insight into the mechanism, by which Akt regulates TAZ.

## Materials and methods

### DNA constructions and virus productions

pGL3 Myo-184- (MyoD-), 8xGT-IIC-δ51LucII- (TEAD-), 9xCAGA-MLP- (SMAD-), p(PRS-1/-4)$_3$- (Pax3-), and pIgk cona-luciferase reporter vectors are gifts of Kenji Miyazawa (Yamanashi University), Hiroshi Sasaki (Kumamoto University), Hiroki Kurihara (The University of Tokyo), and Shoji Yamaoka (Tokyo Medical and Dental University) [51–54]. pCIneoFH-TAZ, pCIneoGFP-TAZ, and pCIneoLuc-TEAD4 were described previously [23, 25].

### Antibodies and reagents

Antibodies and reagents were obtained from commercial sources. The antibodies are listed in the supplementary table (S1 Table). Hoechst 33342, dexamethasone (D4902), epidermal growth factor (E9644), insulin (I5500), and genistein (G6649) (Sigma-Aldrich); anti-DYKDDDDK-tag (014–22383), anti-DYKDDDDK-tag beads (016–22784), basic fibroblast growth factor (064–04541) and Phos-tag acrylamide (WAKO chemicals); LY294002 and Akti-1/2 (Tocris Bioscience); wortmannin and rapamycin (Cayman Chemical); recombinant myostatin (788-G8-010) (R&D systems); and human TNF-α (300-01A) (PeproTech).

### Cell cultures and transfection

HEK293FT cells were cultured in Dulbecco's Modified Eagle Medium (DMEM) containing 4.5 g/l glucose, 584 mg/l glutamine, and 110 mg/l sodium pyruvate (08458–16) (lot: L8H5641) (nacalai tesque) containing 10% fetal bovine serum (FBS) (10270–106) (lot: 42Q4173K) (Thermo Fisher Science) and 10 mM Hepes-NaOH at pH 7.4 under 5% $CO_2$ at 37ºC. MCF10A cells were cultured in DMEM/F12 supplemented with 5% horse serum (Invitrogen), 20 ng/ml epidermal growth factor, 0.5 μg/ml hydrocortisone, and 10 μg/ml insulin. C2C12 cells were passaged in growth medium (DMEM containing 10% FBS) and differentiated in C2C12 differentiation medium (DMEM and 2% horse serum (s0900-500) (lot: S15367S0900) (Biowest)). To induce autophagy, C2C12 cells were cultured in amino acids-deprived DMEM (048–33575) (lot: APN7030) (WAKO chemicals) for 4 h. Primary cultured human skeletal myoblasts (HSkM) were purchased from Thermo Fisher Science. The cells were cultured in low-glucose DMEM (Gibco #11885–084) supplemented with 8% FBS and SkGM SingleQuots Kit (LONZA #CC-41390). To induce differentiation, cells were transferred to the low-glucose DMEM containing 2% horse serum.

### qRT-PCR

mRNA was extracted by use of TRI Reagent® (Molecular Research Center, Inc.). qRT-PCR analysis was performed using SYBR Green (Roche) and ABI7500 Real-Time PCR system (Applied Biosystems) [25]. The primers for mouse genes are; 5'-tacaggccttgctcagctc-3' and 5'-tgtgggagttgcattcactg-3'for *Myog*; 5'-actttctggagccctcctggca-3' and 5'-tttgttgcactacacagcatg-3'for *Myod1*; 5'-ccatggcagtgtcccagccg-3' and 5'-ggcaggcgtgttgacagggg-3' for *Wwtr1*; 5'-tgacctggag-gaaaacattaaga-3' and 5'-agccctgtatgtcttcacactg-3' for connective tissue growth factor (*Ctgf*); 5'-agacctgtgcgccctccgta-3' and 5'-tttgcagcagctcctcgggc-3' for cyclin D1 (*Ccnd1*); 5'-gactcctgca-gagtgaccaag-3' and 5'-cttctacaatgctcttgatgagac-3' for Murf1 *(Trim63)*; 5'-gaatagcatccagatcag-cag-3' and 5'-gagaatgtggcagtgtttgca-3' for Atrogin-1 *(Fbxo32)*; 5'-ccttgaggctcccggcaaat-3' and 5'-actgctccacaaaccaatgga-3' for Musa1 *(Fbxo30)*; 5'-tctcacgctcaaggcaatca-3' and 5'-aagcaaaatc-catcaggcctc-3' for cathepsin L (*Ctsl*); 5'-aactttggcattgtggaagg-3' and 5'-acacattgggggtaggaaca-3' for glyceraldehyde-3-phosphate dehydrogenase (*Gapdh*). We used *Gapdh* as a reference.

## RNA interferences

Mouse TAZ were knocked down in C2C12 cells by use of mouse *Wwtr1* siRNA (s97146) (Ambion) as described previously [25]. Mouse *Akt1* siRNA (SR415524) and mouse *Akt2* siRNA (s62218) were purchased form OriGene and Ambion, respectively.

## Sphere formation assay of MCF10A cells for the initial screening

MCF10A cells were plated at 300 cells/well in 96-well Ultra Low Attachment plates (Corning) and cultured for 10 days in serum-free DMEM/F12 medium containing 5 μg/ml insulin, 10 ng/ml basic fibroblast growth factor, 20 ng/ml epidermal growth factor, and 0.4% (w/v) bovine serum albumin. A cell aggregates larger than 150 μm were defined as spheres.

## Myofusion index

Myofusion index was evaluated as described previously [25]. Briefly, C2C12 cells were immunostained with anti-MHC antibody and nuclei were visualized with Hoechst 33342. The percentage of nuclei included in MHC-positive multinuclear cells was evaluated.

## Cell proliferation assay

Viable cells were evaluated by using MTT colorimetric assay, in which the conversion of 3-(4,5-dimethylthiazol-2-yl)-2,5-diphenyltetrazolium bromide to insoluble tetrazolium by NAD (P)H-dependent cellular oxidoreductase was measured.

## Reporter assay

C2C12 cells were plated at $1 \times 10^5$ cells/well in 12-well plates and cultured overnight. The cells were transfected with pGL3 Myo-184- (MyoD-), 8xGT-IIC-δ51LucII- (TEAD-), 9xCA-GA-MLP- (SMAD-), and p(PRS-1/-4)$_3$- (Pax3-) luciferase reporters alone or with TAZ. In the studies regarding TNF-α and myostatin, HEK293FT cells were plated at $1 \times 10^6$ cells/well in a 6- well plate and transfected with pIgk cona luc (125 ng) and pEF-SEAP (125 ng) or 9xCA-GA-MLP-(SMAD-) luciferase (300 ng) and pCMV alkaline phosphatase (300 ng). 5 h after transfection, cells were harvested and replated at $1 \times 10^5$ cells/well in a 24-well plate. The cells were treated with various combinations of DMSO, 10 ng/ml tumor necrosis factor α, 100 ng/ml myostatin, and 10 μM IBS004735 for 6 h, and then harvested. Luciferase and alkaline phosphatase assays were performed by use of Picagene (Wako Pure Chemical Industries) and CDP-Star (Applied Biosystems), respectively. Luciferase activities were normalized by alkaline phosphatase activities.

## Chromatin immunoprecipitation (ChIP) analysisS

ChIP experiments were performed as described previously [25]. Briefly, C2C12 cells were cross-linked in formaldehyde. After the reaction was quenched for 5 min with 125 mM glycine, cells were lysed in buffer (50 mM Tris-HCl pH 7.5, 150 mM NaCl, 5 mM EDTA, 0.5% (v/v) Nonidet P-40, 1% (v/v) Triton X-100). Chromatin was sheared by 25 consecutive rounds in a sonicator bath and was incubated with 2 μg of antibodies. Immunoprecipitation was performed with 20 μl of protein G-Sepharose beads. Protein G-Sepharose with no antibody was used as the control (mock ChIP). Immunoprecipitated DNA fragments were isolated with Chelex-100 resin and diluted 1:2.5 for quantitative PCR analysis.

## Subcellular fractionation

C2C12 cells were plated at $4 \times 10^6$ cells/10-cm plate. Cells were transferred to the differentiation medium, cultured with DMSO or 10 mM IBS004735 for 24 h, and harvested in 1 ml ice-cold PBS. Cells were collected by centrifugation at 200 x g for 7 min and homogenized in 0.5 ml STM buffer (50 mM Tris-HCl pH 7.4, 250 mM sucrose, and 5 mM $MgCl_2$) by 50 strokes in a glass homogenizer. The samples were kept on ice for 30 min and then vortexed for 15 sec. 30 μl of the sample was saved as a whole cell lysate. The remaining sample was centrifuged at 800 x g for 15 min to separate the first supernatant and the first pellet. The supernatant was centrifuged at 800 x g for 10 min again to collect the second supernatant, which was centrifuged at 11,000 x g for 10 min. The third supernatant was used as a cytosolic fraction. The first pellet was resuspended in 470 μl STM buffer and centrifuged at 500 x g for 15 min. The second pellet was resuspended in 470 μl STM buffer and centrifuged at 1,000 x g for 15 min. The third pellet was resuspended in 470 μl NET buffer (20 mM Hepes-NaOH pH 7.9, 500 mM NaCl, 0.2 mM EDTA, 1.5 mM MgCl2, 20% glycerol, and 1% (w/v) TritonX-100), kept on ice for 30 min, vortexed for 15 sec, homogenized by sonication, and centrifuged at 9,000 x g for 30 min to collect the supernatant, which was used as a nuclear fraction.

## Phosphate-affinity SDS-PAGE

Phosphate-affinity SDS-PAGE was performed by use of Phos-tag™ acrylamide (Wako chemicals) and PolyVinylidene DiFluoride (PVDF) membranes as described previously [23].

## *In vitro* SUnSET experiment

C2C12 cells were cultured under the differentiation condition with DMSO or 10 μM IBS004735 for the indicated periods of time. 1 μM puromycin was added to the medium. 30 min later, the cells were harvested and the incorporated puromycin was detected with anti-puromycin antibody.

## Synthesis of derivatives of IBS004735

Typical procedures of synthesis of derivatives are described in **S3 Fig**. Other derivatives shown in **S4 Fig** were synthesized by similar methods, and the structures and purities were confirmed by [1]H NMR.

## Statistical analysis

Statistical analyses were performed with student's t test for the comparison between two samples and analysis of variance (ANOVA) with Dunnett's test for the multiple comparison using the GraphPad Prism 5.0 (GraphPad Software).

## Other procedures

Immunoprecipitation and immunofluorescence were performed as described previously [55].

## Supporting information

**S1 Fig. Summary of the initial screening of TAZ activators that promote myogenesis in C2C12 cells.**
(PDF)

**S2 Fig. Structures of fifty TAZ activators obtained through the initial screening.**
(PDF)

**S3 Fig. Synthesis of derivatives of IBS004735.**
(PDF)

**S4 Fig. Derivatives of IBS00473 and their effect.**
(PDF)

**S5 Fig. Uncropped data for Fig 1.**
(PDF)

**S6 Fig. Uncropped data for Fig 3.**
(PDF)

**S7 Fig. Uncropped data for Fig 4A and Fig 4B.**
(PDF)

**S8 Fig. Uncropped data for Fig 4C and Fig 4F.**
(PDF)

**S9 Fig. Uncropped data for Fig 5 and Fig 6.**
(PDF)

**S10 Fig. Uncropped data for Fig 7.**
(PDF)

**S1 Table. The list of the antibodies.**
(DOCX)

## Acknowledgments

We are grateful for Hiroshi Asahara (Tokyo Medical and Dental University), Yasutomi Kamei (Kyoto Prefectural University), Hiroki Kurihara (The University of Tokyo), Yoshihiro Ogawa (Tokyo Medical and Dental University), Hiroshi Sasaki (Kumamoto University), Kenji Miyazawa (Yamanashi University), Kohei Miyazono (The University of Tokyo), Hiroshi Takayanagi (The University of Tokyo), Sumiko Watanabe (The University of Tokyo), and Shigeru Yamada (Jissen Women's University) for the materials and advice. X.X. and Z.Y were supported by Japanese Government (Monbukagakusho) (MEXT) scholarship. K.I. is a Research Fellow of Japan Society for the Promotion of Science (JSPS Research Fellow).

## Author Contributions

**Conceptualization:** Kyoko Arimoto-Matsuzaki, Yutaka Hata.

**Data curation:** Manami Kodaka.

**Funding acquisition:** Kyoko Arimoto-Matsuzaki, Yutaka Hata.

**Investigation:** Manami Kodaka, Fengju Mao, Masami Kitamura, Xiaoyin Xu, Zeyu Yang.

**Methodology:** Takuya Oyaizu, Naoki Yamamoto.

**Resources:** Mari Ishigami-Yuasa, Nozomi Tsuemoto, Shigeru Ito, Hiroyuki Kagechika.

**Supervision:** Kyoko Arimoto-Matsuzaki, Kentaro Nakagawa, Junichi Maruyama, Kana Ishii, Chihiro Akazawa, Takuya Oyaizu, Naoki Yamamoto, Hiroshi Nishina.

**Writing – original draft:** Manami Kodaka, Kyoko Arimoto-Matsuzaki, Yutaka Hata.

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
