## [Decision Letter · Decision Letter 0]

2 Dec 2019

PONE-D-19-31614

Characterization of a novel compound that promotes myogenesis via Akt and transcriptional co-activator with PDZ-binding motif (TAZ) in mouse C2C12 cells

PLOS ONE

Dear Dr. Hata,

Thank you for submitting your manuscript to PLOS ONE. After careful consideration, we feel that it has merit but does not fully meet PLOS ONE’s publication criteria as it currently stands. Therefore, we invite you to submit a revised version of the manuscript that addresses the points raised during the review process.

As noted below, both the reviewers noted a few minor deficiencies and clarifications, and requested a few additional experiments possibly inclusion of a key validation experiment utilizing primary human myoblasts.  

We would appreciate receiving your revised manuscript by Jan 16 2020 11:59PM. To enhance the reproducibility of your results, we recommend that if applicable you deposit your laboratory protocols in protocols.io, where a protocol can be assigned its own identifier (DOI) such that it can be cited independently in the future. For instructions see: http://journals.plos.org/plosone/s/submission-guidelines#loc-laboratory-protocols

We look forward to receiving your revised manuscript.

Kind regards,

Arun Rishi, Ph.D.

Academic Editor

PLOS ONE

Journal Requirements:

'Yutaka Hata collaborated with Shinogi Co. Ltd. to develop drugs against sarcopenia between 2017 and 2018.'

Additional Editor Comments (if provided):

Reviewers' comments:

Reviewer's Responses to Questions

**Comments to the Author**

1. Is the manuscript technically sound, and do the data support the conclusions?

Reviewer #1: Yes

Reviewer #2: Yes

2. Has the statistical analysis been performed appropriately and rigorously? 

Reviewer #1: N/A

Reviewer #2: Yes

3. Have the authors made all data underlying the findings in their manuscript fully available?

Reviewer #1: Yes

Reviewer #2: Yes

4. Is the manuscript presented in an intelligible fashion and written in standard English?

Reviewer #1: Yes

Reviewer #2: Yes

5. Review Comments to the Author

Reviewer #1: The authors of this manuscript identified a novel activator compound of TAZ and promotes myogenesis via Akt.This manuscript is organized well and data could support the conclusion. There are some points which could be improved.

1. In figure 1D, in IBS004735 treatment groups, protein expression levels of TAZ and Myogenin were elevated significantly in 48h but decreased in 72h, how to explain this phenomenon?

2. In figure 1E, the TAZ siRNA transfection efficiency detected by WB should be repeated.

3. In figure 4F, either in control siRNA or in TAZ siRNA group, protein expression level of TAZ decreased under IBS004735 treatment with 48h, this seemed to contradict to the result in fig 1D.

4. The introduction needs to be reorganized.

Reviewer #2: Kodaka and colleagues investigated the effects of IBS004735 on TAZ activation in cultured C2C12 cells. The main finding is that IBS004735 stimulates TAZ activity via Akt/mTORC1 signaling. The experiments are conducted with a high technical standard. Overall, the manuscript is well written and contains sufficient details for the readers. The major limitation of the work is that all experiments are performed in C2C12 cell culture, and therefore, the relevance of the findings in humans remains unknown. This limitation could be addressed by repeating key experiments in primary human myoblasts. Otherwise, I only have a few comments.

• The compound IBS004735 was selected as one among many compounds known to promote myogenesis in C2C12 cells. It would be helpful for the reader to understand why the authors selected exactly this compound and not another one. I suggest that this information is included in the introduction.

• Line 101-116 summarized previous work and introduce the rationale for the present investigation. I fully recognize the need for this information, but the authors should consider moving such information to the introduction to avoid any repetitions.

• Figure 4A: please specify the specific phosphorylation sites at Akt, mTOR, and S6K.

• Figure 6F: please specify the specific phosphorylation sites at ULK1. Some sites are activating and some are inhibiting ULK1 activity.

• PI3K activity is usually regulated by upstream enzymes or receptors that mediates the effect of growth factors and hormones. How does IBS004735 regulated PI3K activity?

• I suggest that the most important experiments are reproduced in primary human myoblasts to check if the findings can be translated to humans.

6. PLOS authors have the option to publish the peer review history of their article (what does this mean?). If published, this will include your full peer review and any attached files.

Reviewer #1: No

Reviewer #2: Yes: Andreas Buch Møller

---

## [Author Response · Author response to Decision Letter 0]

6 Mar 2020

Manuscript Number: PONE-D-19-31614

Title: Characterization of a novel compound that promotes myogenesis via Akt and transcriptional co-activator with PDZ-binding motif (TAZ) in mouse C2C12 cells

Authors: Kodaka M, Mao F, et al 

We thank reviewers for very useful comments and have thoroughly revised the manuscript according to their comments. Our responses are listed as follows.

Reponses to the comments of Reviewer #1

The authors of this manuscript identified a novel activator compound of TAZ and promotes myogenesis via Akt. This manuscript is organized well and data could support the conclusion. There are some points which could be improved.

We greatly appreciate the comments. We are very happy to hear that our conclusion is basically supported by our findings.

1. In figure 1D, in IBS004735 treatment groups, protein expression levels of TAZ and Myogenin were elevated significantly in 48h but decreased in 72h, how to explain this phenomenon?

We greatly appreciate this comment. As the reviewer pointed out, the finding shown in the original figure was incongruous. The reviewer’s comment prompted us to repeat the experiment and re-evaluated the protein expression. In conclusion, we found that we need to change our argument, TAZ and myogenin are not reduced at 72 h and remain at almost the same level as at 48 h. We have replaced Figure 1D with a new one, and have revised the text too (Figure 1D; page 4, line 113).

2. In figure 1E, the TAZ siRNA transfection efficiency detected by WB should be repeated.

We reperformed the experiment shown in Figure 1E and replaced W-B with a new one (Figure 1F). We hope the current figure clearly demonstrates that TAZ is efficiently suppressed.

3. In figure 4F, either in control siRNA or in TAZ siRNA group, protein expression level of TAZ decreased under IBS004735 treatment with 48h, this seemed to contradict to the result in fig 1D.

As we did for Figure 1D, we repeated the experiment for Figure 4F and revaluated TAZ expression. Under the transfection with control siRNA, TAZ expression did not change as in Figure 1D. We replaced the image with a new one (Figure 4F, siCont, TAZ). In terms of TAZ siRNA group, TAZ was suppressed as expected in Figure 4F, while in Figure 1D, we did not perform the silencing of TAZ. Therefore, we consider that there was no contradiction here. As we repeated the experiment, we replaced the image with a new one in Figure 4F, siTAZ, TAZ. We hope that we have properly responded to the reviewer’s comment.

4. The introduction needs to be reorganized.

As Reviewer #2 also requested us to revise the introduction, we reorganized Introduction and summarized the original screening to explain why we focused on IBS004735 in this study (page 4, line 85 to line 105).

Reponses to the comments of Reviewer #2

Kodaka and colleagues investigated the effects of IBS004735 on TAZ activation in cultured C2C12 cells. The main finding is that IBS004735 stimulates TAZ activity via Akt/mTORC1 signaling. The experiments are conducted with a high technical standard. Overall, the manuscript is well written and contains sufficient details for the readers. The major limitation of the work is that all experiments are performed in C2C12 cell culture, and therefore, the relevance of the findings in humans remains unknown. This limitation could be addressed by repeating key experiments

in primary human myoblasts.

We greatly appreciate this comment. Encouraged by this comment, we applied our compound to primary human myoblasts and confirmed its activity. We added the information in Materials and Methods and described the results (Figure 7; page 7, line 223 to line 228; page 9, line 323 to 327).

Otherwise, I only have a few comments.

• The compound IBS004735 was selected as one among many compounds known to promote myogenesis in C2C12 cells. It would be helpful for the reader to understand why the authors selected exactly this compound and not another one. I suggest that this information is included in the introduction.

• Line 101-116 summarized previous work and introduce the rationale for the present investigation. I fully recognize the need for this information, but the authors should consider moving such information to the introduction to avoid any repetitions.

According these two comments and the comment of Reviewer #1, we have reorganized Introduction (page 4, line 85 to line 105). We hope that readers can follow why we focused on IBS004735 in this study.

• Figure 4A: please specify the specific phosphorylation sites at Akt, mTOR, and S6K.

We have described the specific phosphorylation sites at Akt, mTOR, and S6K in the revised Figure 4A.

•Figure 6F: please specify the specific phosphorylation sites at ULK1. Some sites are activating and some are inhibiting ULK1 activity.

We have described the specific phosphorylation site at ULK1 and described that this site is an activating phosphorylation site (Figure 6; page 7, line 220 and line 221).

• PI3K activity is usually regulated by upstream enzymes or receptors that mediates the effect of

growth factors and hormones. How does IBS004735 regulated PI3K activity?

As shown in Figure 5B, tyrosine kinase inhibitor blocks the effect of IBS004735. Therefore, we speculate that IBS004735 activates a certain tyrosine kinase receptor. Actually, we attempted during the revision to identify which tyrosine kinase is activated by IBS004735. As far as we know, IBS004735 does not activate Insulin receptor, IGF-1 receptor, PDGF receptor, EGF receptors, ErB2, ErB4, VEGF receptor, Met, or ALK. We expressed these receptors in HEK293 cells and treated the cells with IBS004735. However, to our regret, none of them was activated by IBS004735. A recent study also discusses that various receptor tyrosine kinases activate TAZ. In the revised manuscript, we have cited this paper and have added discussion (page 8, line 272 to line 274).

• I suggest that the most important experiments are reproduced in primary human myoblasts to check if the findings can be translated to humans.

Thank you very much for your suggestion. In the revised manuscript, we can show that IBS004735 works not only on C2C12 cells but also on primary cultured myoblasts. We have described the result and the method (Figure 7; page 7, line 223 to line 228; page 9, line 323 to 327)

Other changes

1. Fengju Mao performed experiments for the revision. Thus, in the revised manuscript, it is described that Kodaka and Mao equally contributed to this work (page 1, line 32).

2. In the original manuscript, we commented “ethacridine does not promote myogenesis in C2C12 cells”. But this argument is not directly related to this study. Hence, we have deleted this statement and have omitted the phrase “data not shown” according to the policy of PLOS ONE.

3. We have added the statement “This does not alter our adherence to PLOS ONE policies on sharing data and materials” to COI (page 12, line 438).

4. We have added the captions for Supporting information at the end of the manuscript (page 12, line 440 to page 13, line 452).

5. We have shown uncropped blot/gel images in Supplementary information (Supplementary Figure 5 to Supplementary Figure 10).

---

## [Decision Letter · Decision Letter 1]

20 Mar 2020

Characterization of a novel compound that promotes myogenesis via Akt and transcriptional co-activator with PDZ-binding motif (TAZ) in mouse C2C12 cells

PONE-D-19-31614R1

Dear Dr. Hata,

We are pleased to inform you that your manuscript has been judged scientifically suitable for publication and will be formally accepted for publication once it complies with all outstanding technical requirements.

With kind regards,

Arun Rishi, Ph.D.

Academic Editor

PLOS ONE

Additional Editor Comments (optional):

Reviewers' comments:

Reviewer's Responses to Questions

**Comments to the Author**

1. If the authors have adequately addressed your comments raised in a previous round of review and you feel that this manuscript is now acceptable for publication, you may indicate that here to bypass the “Comments to the Author” section, enter your conflict of interest statement in the “Confidential to Editor” section, and submit your "Accept" recommendation.

Reviewer #2: All comments have been addressed

2. Is the manuscript technically sound, and do the data support the conclusions?

Reviewer #2: Yes

3. Has the statistical analysis been performed appropriately and rigorously? 

Reviewer #2: Yes

4. Have the authors made all data underlying the findings in their manuscript fully available?

Reviewer #2: Yes

5. Is the manuscript presented in an intelligible fashion and written in standard English?

Reviewer #2: Yes

6. Review Comments to the Author

Reviewer #2: All comments have been satisfactory addressed and the revised manuscript did not raise further concerns.

7. PLOS authors have the option to publish the peer review history of their article (what does this mean?). If published, this will include your full peer review and any attached files.

Reviewer #2: Yes: Andreas Buch Møller

---

## [Editor Report · Acceptance letter]

24 Mar 2020

PONE-D-19-31614R1 

Characterization of a novel compound that promotes myogenesis *via* Akt and transcriptional co-activator with PDZ-binding motif (TAZ) in mouse C2C12 cells 

Dear Dr. Hata:

I am pleased to inform you that your manuscript has been deemed suitable for publication in PLOS ONE. Congratulations! Your manuscript is now with our production department. 

With kind regards,

on behalf of

Prof Arun Rishi 

Academic Editor

PLOS ONE